# Aggregation of Genome-Wide Association Data from FinnGen and UK Biobank Replicates Multiple Risk Loci for Pregnancy Complications

**DOI:** 10.3390/genes13122255

**Published:** 2022-11-30

**Authors:** Anton I. Changalidis, Evgeniia M. Maksiutenko, Yury A. Barbitoff, Alexander A. Tkachenko, Elena S. Vashukova, Olga V. Pachuliia, Yulia A. Nasykhova, Andrey S. Glotov

**Affiliations:** 1Dpt. of Genomic Medicine, D.O. Ott Research Institute of Obstetrics, Gynaecology, and Reproductology, 199034 St. Petersburg, Russia; 2Faculty of Software Engineering and Computer Systems, ITMO University, 197101 St. Petersburg, Russia; 3Dpt. of Genetics and Biotechnology, St. Petersburg State University, 199034 St. Petersburg, Russia

**Keywords:** pregnancy complications, GWAS, preeclampsia, hyperemesis gravidarum, gestational diabetes, preterm birth, genetic associations

## Abstract

Complications endangering mother or fetus affect around one in seven pregnant women. Investigation of the genetic susceptibility to such diseases is of high importance for better understanding of the disease biology as well as for prediction of individual risk. In this study, we collected and analyzed GWAS summary statistics from the FinnGen cohort and UK Biobank for 24 pregnancy complications. In FinnGen, we identified 11 loci associated with pregnancy hypertension, excessive vomiting, and gestational diabetes. When UK Biobank and FinnGen data were combined, we discovered six loci reaching genome-wide significance in the meta-analysis. These include rs35954793 in *FGF5* ($p=6.1\times 10^{-9}$), rs10882398 in *PLCE1* ($p=8.9\times 10^{-9}$), and rs167479 in *RGL3* ($p=5.2\times 10^{-9}$) for pregnancy hypertension, rs10830963 in *MTNR1B* ($p=4.5\times 10^{-41}$) and rs36090025 in *TCF7L2* ($p=3.4\times 10^{-15}$) for gestational diabetes, and rs2963457 in the *EBF1* locus ($p=6.5\times 10^{-9}$) for preterm birth. In addition to the identified genome-wide associations, we also replicated 14 out of 40 previously reported GWAS markers for pregnancy complications, including four more preeclampsia-related variants. Finally, annotation of the GWAS results identified a causal relationship between gene expression in the cervix and gestational hypertension, as well as both known and previously uncharacterized genetic correlations between pregnancy complications and other traits. These results suggest new prospects for research into the etiology and pathogenesis of pregnancy complications, as well as early risk prediction for these disorders.

## 1. Introduction

Pregnancy complications are a wide range of pathologies that affect about 15% of all pregnancies. The most serious complications include hypertensive disorders of pregnancy, including preeclampsia, and gestational diabetes which could lead to impaired growth and development of the fetus, placental abruption, and preterm labor [1,2,3]. Hypertensive disorders of pregnancy affect almost 10% of gestations [4], with pre-eclampsia incidence at around 5–8% [5]. Gestational diabetes incidence varies between 2 and 38% in different populations (according to March 2021 data from https://diabetesatlas.org/data/en/indicators/14/ (accessed on 5 November 2022)). In Russia, hypertension complicates 5–30% of pregnancies (according to the 2014 data of the Ministry of Healthcare of Russia). GD frequency in Russia is estimated at 2–9% [6,7]. According to the statistics, hypertensive disorders of pregnancy, the second leading cause of maternal mortality worldwide, account for 14% of all maternal deaths [8]. Previously it has been shown that pregnancy-specific complications continue to affect maternal health long after baby delivery and could lead to increased risk of vascular disease in later life. Gestational diabetes mellitus (GD) is an independent risk factor for future type 2 diabetes (T2D) mellitus, metabolic syndrome, cardiovascular morbidity, vascular endothelial dysfunction, renal and ophthalmic disease. The risk of these conditions may be decreased with proper prevention and interventions [9]. Unfortunately, the etiology and pathogenesis of pregnancy-related disorders is not fully understood, several recent studies have suggested that it could be the result of a combination of genetic, epigenetic, and environmental factors [1]. Currently, prevention treatment of pregnancy complications is hindered by the lack of robust early predictors.

Genome-wide association studies (GWAS) have been widely used to identify the genomic regions on chromosomes that harbor genetic determinants of complex traits [10]. A limited number of studies have so far identified several dozen single nucleotide polymorphisms (SNPs) in different genes associated with susceptibility to pregnancy complications in various ethnic groups. For example, several hypertension-related variants, including rs1458038 in *FGF5*, rs1918975 in *MECOM*, and rs10774624 in *SH2B3*, were found to be significantly associated with the risk of preeclampsia [11]. In another study of a Korean population, two loci were found to be associated with gestational diabetes, including rs10830962 in *MTNR1B* and rs7754840 in *CDKAL1* [12]. A recent GWAS for preterm birth and gestational duration in European women identified three risk loci associated with spontaneous preterm birth, spanning the *EBF1*, *EEFSEG*, and *AGTR2* genes [13].

Previously, we have used publicly available genome-wide association data from the UK Biobank (UKB) to describe 4 genome-wide significant loci associated with several pregnancy complications [14]. These included a region spanning the novel *FBLN7* gene associated with pregnancy hypertension. In this study, we expanded our analysis by including another publicly available resource, FinnGen (FG), and investigated genome-wide association data for 24 different pregnancy complications. Besides, we conducted additional annotation of the results, and replicated multiple known genetic associations for pregnancy-related disorders.

## 2. Materials and Methods

### 2.1. Data Collection

We used GWAS summary statistics from 2 publicly available biobank-scale cohorts: UK Biobank (UKB) [15] and FinnGen (FG) [16]. UK Biobank is a prospective population-based project primarily including UK residents aged 40–69 years. FinnGen is a study that, in contrast, includes both population-based and disease-oriented cohorts. We selected matching pregnancy-related disorders from UKB and FG using keyword-based search and manual matching of traits. The list of traits included in the analysis is shown in Appendix A.

The final set of 24 traits includes three distinct diagnoses that correspond to hypertensive disorders of pregnancy. In addition to preeclampsia, the dataset includes (i) I9_HYPTENSPREG (Hypertension complicating pregnancy, childbirth, and the puerperium), termed HP in the text, which includes all types of hypertensive disorders, both pre-existing and emerging for the first time during pregnancy; and (ii) gestational (pregnancy-induced) hypertension without significant proteinuria (denoted O13 in the UKB data and O15_GESTAT_HYPERT in the FG data), termed GH in the test, which corresponds to hypertension that emerges for the first time during pregnancy. In contrast to both HP and GH, preeclampsia is diagnosed only when other symptoms are present besides hypertension, with significant proteinuria being the main hallmark of PE. In accordance with this aforementioned assumption, HP GWAS has the greatest number of cases among all other hypertensive disorders (1.78 times greater than GH in FG, and 2.83 times greater in UKB). Similarly, the sample overlap between the traits is 55.96% between HP and GH and 14.57% between GH and PE (according to the FG r6 data).

Pre-computed summary statistics of round 2 UKB GWAS were obtained from the Neale lab website (http://www.nealelab.is/uk-biobank (accessed on 25 February 2022)). FinnGen release 6 data was collected from the project website. Details regarding raw data analysis methods can be found at https://github.com/Nealelab/UK_Biobank_GWAS#imputed-v3-association-model (accessed on 25 February 2022) for UKB and https://finngen.gitbook.io/documentation/methods/phewas (accessed on 25 February 2022) for FG. Briefly, UKB GWAS summary statistics are generated using the linear regression model with the following covariates: first 20 PCs, age, and squared age. FG uses the mixed model regression approach with sex, age, first 10 PCs, and genotyping batch used as covariates.

### 2.2. GWAS Meta-Analysis

As the FinnGen project, in contrast to UK Biobank, uses GRCh38 reference assembly, we first mapped the FG summary statistics from GRCh38 to GRCh37 using the UCSC liftOver tool [17]. We also excluded low-confidence variants and filtered the data by minor allele frequency (variants with MAF >0.01 were considered for meta-analysis, and MAF >0.05 cutoff was used to select significant genome-wide associations).

We used 2 different software tools for meta-analysis: METAL [18] and MTAG (Multi-Trait Analysis of GWAS) [19]. A sample size-based model in METAL was used to avoid confounding by data preprocessing methods used by UKB or FG. The results of METAL and MTAG showed perfect correlation; hence, METAL-based *p*-values are shown in the text. Lead SNPs at loci were selected as the ones with the best *p*-value. Annotation of lead SNP genes was performed using the NCBI eUtils toolkit.

### 2.3. Functional Annotation and Genetic Correlation Analysis

Functional annotation of GWAS summary statistics was performed using the FUMA tool [20]. The parameters used for FUMA analysis can be found in the Appendix A (most parameters were set to default values). The genome-wide genetic correlation analysis was performed using the LDAK toolkit [21]. As a reference, we used a collection of 612 pairwise meta-analysis results for traits from UKB and FG. To construct such a collection, phenotype text descriptions were transformed into vector representation using the stsb-roberta-large model, phenotypes were matched based on cosine similarity between vector representations, and the resulting pairs of phenotypes were manually curated to exclude false matches.

Point estimate and the standard deviation of the genetic correlation coefficient ($r_{g}$) were used to calculate the statistical significance using a one-parameter Wald test. Multiple testing corrections were achieved using the standard Bonferroni method. Modified code from Ganna et al. [22] and COVID-19 Host Genetics Initiative [23] was used for forest plot and heatmap generation. If a point estimate of the correlation or parts of confidence interval exceeded the range between −1 and 1, the estimate and/or confidence interval margin were set to −1 or 1, respectively.

For statistical fine-mapping of the association signal in meta-analysis, we used FINEMAP [24] v1.4.1 and the 1000 Genomes [25] European reference genotype panel. The maximum size of the credible set was set to 5, and SNPs within the 3 Mb window around the lead SNP were included. For FG, fine-mapping results were acquired from the FG data repository. For each locus, we further considered all SNPs with posterior inclusion probability ≥0.2.

### 2.4. Data Availability

All data and code pertinent to the analysis presented in this work are available in the GitHub repository: https://github.com/bioinf/pregnancy_meta_analysis.

## 3. Results

### 3.1. Characterization of Significant Associations for Pregnancy Complications

UK Biobank (UKB) [15] and FinnGen (FG) [16] are two of the largest publicly available resources that provide GWAS summary statistics for thousands of traits in individuals of European ancestry. Previously, we have collected and described genome-wide associations with pregnancy complications in the UKB [14], with four loci identified as significant genome-wide associations. In the first stage of this work, we collected the data for a similar set of 24 pregnancy complications from FG.

The set of traits is available in Appendix A (see Methods for more details on phenotype selection and definitions). Generally, the FG sample contained several times more cases than UKB for all traits (Appendix A), suggesting that FG GWAS should boast greater statistical power to identify significantly associated loci. An initial assessment of the data identified four traits with significant genome-wide associations in FG. These included (i) hypertension complicating pregnancy, childbirth, and the puerperium (HP); (ii) gestational hypertension (GH); (iii) excessive vomiting in pregnancy (EV) (likely representing hyperemesis gravidarum (HG)); and (iv) gestational diabetes (GD) (Figure 1). Overall, we identified 11 genome-wide significant loci (Figure 1, Table 1): 5 for HP, 1—for GH, 1—for EV, and 4—for GD.

Loci associated with HP included both well-known genes linked to pregnancy complications, such as *MTHFR*, *FGF5*, and *ZNF831*, as well as novel ones—*PLCE1* and *RGL3*. For GH, we discovered one genome-wide significant locus corresponding to the novel *PREX1* gene. The loci identified in the GD GWAS also included known genes affecting both type 2 diabetes and GD—*GCKR*, *MTNR1B*, and *TCF7L2*, as well as the MHC region. The locus identified in the excessive vomiting GWAS corresponded to the *GDF15* gene, which has recently been found to be associated with HG [27]. We also explored the results of statistical fine-mapping of the GWAS signal at the identified loci to identify potentially causal variant sets (see Materials and Methods for details). The results of fine-mapping (Appendix A) showed that the vast majority of the potentially causal variants were outside protein-coding regions. Of note, we observed an inclusion of the rs1260326 missense variant in *GCKR* in the credible set for the GD locus on chromosome 2, providing further confirmation of the *GCKR* role in the disease.

Next, we attempted to perform meta-analysis of UKB and FG data for the 24 traits used in our initial analysis. This analysis was performed using the METAL [18] tool and the sample size-based statistical model (see Materials and Methods for more details). Among the 24 traits, we discovered genome-wide significant loci in only three meta-analyses—for HP, GD, and preterm birth (PTB) (Figure 2, Table 2). It is noteworthy that, while the genome-wide associations were present in the FG data for HP and GD, no significant hits for PTB were present in either UKB or FG. Hence, the associations for PTB represent loci which can only be identified in the combined sample.

In total, 6 loci were identified as genome-wide significant hits in the meta-analysis: 3—for HP, 2—for GD, and 1—for PTB (Table 2). All GD and HP associations were also identified in the FG-only analysis described above. For HP, significant meta-analysis findings corresponded to the *FGF5*, *PLCE1*, and *RGL3* genes (Figure 2a). The two loci identified in the GD meta-analysis corresponded to the *MTNR1B* and *TCF7L2* genes (Figure 2b). Another finding corresponded to the locus associated with PTB (Figure 2c). The lead SNP, rs2963457, is located in a wide intergenic region on chromosome 5, with the closest genes, *CLINT1* and *EBF1*, located 621.8 kbp upstream and 215.0 kbp downstream of the variant consequently. The most likely causal gene in the locus is *EBF1*, which has previously been implicated in PTB [28]. The rs2963457 variant identified in our analysis is in high linkage with the rs2963463 variant, previously identified as the top variant in PTB GWAS [13]. Similarly to FG-based analysis, we performed statistical fine-mapping of the GWAS signal at the identified loci (Appendix A). However, no notable coding variants were identified as potentially causal by fine-mapping.

### 3.2. Annotation of GWAS Results for Pregnancy Complications

We next performed functional annotation of the results for the four aforementioned GWAS datasets. To this end, we used the FUMA toolkit [20]. Analysis of FG data using FUMA identified a significant association between genes specifically expressed in the cervix and endocervix and risk loci for HP and GH. In addition, genes expressed in the blood vessel were also enriched at GH risk loci (Appendix A). FUMA did not identify significant (FDR-adjusted p-value<0.05) enrichment of curated gene sets at the identified loci. However, several top-ranking gene sets (uncorrected $p\text{-}\mathrm{value}<5\times 10^{-5}$) deserve to be mentioned. These include microtubule binding proteins enriched at HP and GH loci, genes involved in vasculogenesis with enrichment at HP loci, and embryonic cleavage genes modestly overrepresented at EV loci (full results are available in Appendix A). Functional annotation of meta-analysis summary statistics using FUMA did not identify any significant pathway- or tissue-level associations (complete FUMA results for all traits are available as Appendix A).

Gene-based analysis in FUMA also confirmed the role of major genes at the loci identified in both FG and UKB+FG meta-analysis, such as *MTHFR* for HP and GH, *MTNR11B* and *TCF7L2* for GD. In addition, FUMA highlighted two additional genes: *PRRT1* for GD (in meta-analysis), and *MPHOSPH10* for GH (in FG data).

We next went on to conduct other types of downstream analysis for the obtained GWAS results. We first analyzed genome-wide genetic correlation within the FG dataset. To do so, we tested the correlation between the 4 traits with significant genome-wide associations (HP, GH, EV, and GD) and the remaining set of 1187 traits in FG. In total, we identified 62 significant pairwise genetic correlations. These included both correlations supported by other studies (Figure 3a) and previously uncharacterized connections (Figure 3b). In the first category, we identified significant correlation between HP/GH and various forms of hypertension and cardiological diseases, as well as between GD and such traits as diabetes mellitus (DM) ($r_{g}=1\pm 0.28$, $p\text{-}\mathrm{value}=1.14\times 10^{-8}$) and soft tissue disorders ($r_{g}=0.57\pm 0.21$, $p\text{-}\mathrm{value}=9.95\times 10^{-8}$). Interestingly, we did not observe a significant correlation between GD and obesity, though both HP and GH showed significant association with obesity ($r_{g}=0.49\pm 0.16$, $p\text{-}\mathrm{value}=1.23\times 10^{-9}$; rg=0.42±0.16, $p\text{-}\mathrm{value}=1.28\times 10^{-7}$). In the latter category (Figure 3b) we discovered several intriguing correlations for GD which included polyarthropathies ($r_{g}=0.46\pm 0.20$, $p\text{-}\mathrm{value}=7.48\times 10^{-6}$), nerve root diseases ($r_{g}=0.85\pm 0.30$, $p\text{-}\mathrm{value}=3.19\times 10^{-8}$), diseases of the ear ($r_{g}=0.64\pm 0.24$, $p\text{-}\mathrm{value}=2.45\times 10^{-7}$), and gastrointestinal pathologies ($r_{g}=0.51\pm 0.24$, $p\text{-}\mathrm{value}=3.94\times 10^{-5}$). In addition, we identified a correlation between GH and dizziness/giddiness ($r_{g}=0.69\pm 0.33$, $p\text{-}\mathrm{value}=3.69\times 10^{-5}$), and between HP and nerve root diseases ($r_{g}=0.46\pm 0.19$, $p\text{-}\mathrm{value}=1.4\times 10^{-6}$).

For meta-analysis results, we calculated genetic correlation between HP, GD, PTB and a collection of 612 pairwise meta-analysis results. This analysis identified 6 pairs of traits with significant genome-wide genetic correlation (Figure 3c). Of these, five diseases showed significant genetic correlation only with HP: ischemic heart disease ($r_{g}=0.45\pm 0.16$, $p\text{-}\mathrm{value}=6.7\times 10^{-8}$), unspecified maternal hypertension ($r_{g}=0.78\pm 0.35$, $p\text{-}\mathrm{value}=1.1\times 10^{-5}$), primary (essential) hypertension ($r_{g}=0.70\pm 0.20$, $p\text{-}\mathrm{value}=5.4\times 10^{-12}$), obesity ($r_{g}=0.37\pm 0.18$, $p\text{-}\mathrm{value}=5.6\times 10^{-5}$), and pain in throat and chest ($r_{g}=0.36\pm 0.15$, $p\text{-}\mathrm{value}=5.3\times 10^{-6}$). At the same time, diabetes mellitus (DM) correlated with both GD ($r_{g}=0.93\pm 0.32$, $p\text{-}\mathrm{value}=1.7\times 10^{-8}$) and HP ($r_{g}=0.42\pm 0.19$, $p\text{-}\mathrm{value}=2.1\times 10^{-5}$).

Taken together, the results of the genetic correlation analysis show common genetics behind essential (primary) hypertension and pregnancy hypertension, as well as between diabetes, GD, and HP. Importantly, none of the three loci identified in the HP meta-analysis showed genome-wide significance in the essential hypertension or maternal hypertension meta-analysis. This suggests that these loci may be, at least to a certain degree, specific to pregnancy hypertension. Genetic correlation analysis also revealed interesting novel correlations for GD, which may require further investigation.

### 3.3. Replication of Known Associations in the FG and UKB Data

Having identified the genome-wide significant loci for pregnancy complications in FG and combined FG + UKB data, we next sought to extend our analysis by attempting to replicate other reported associations for the studied traits. To this end, we created a list of known variants associated with pregnancy complications by combining association data from the GWAS Catalog and several studies on PE [11], HG [27], PTB [13,29,30,31], GD [12], miscarriage [32], and prolonged gestation [33,34]. Only variants reaching genome-wide significance were included in the analysis. In total, 40 associations were selected for replication: 11—for hyperemesis gravidarum, 2—for GD, 6—for PE, and 14—for PTB, and 7—for other traits (Appendix A). Replication of associations for hyperemesis gravidarum and PE was performed using meta-analysis results for EV and HP, respectively.

14 out of 40 associations were successfully replicated in our results. Of these, 12 variants showed significant association in both FG and combined FG + UKB analysis, while the remaining variants were only significant in the FG cohort. Seven of the replicated variants corresponded to loci that were not identified as genome-wide significant (Table 3): 2—for HG, 1—for GD, and 4—for PE. Of note, we did not replicate any of the reported associations for prolonged gestation, spontaneous abortion, and premature rupture of membranes. Among successfully replicated variants, the rs143409503 and rs2508362 variants in the *IGFBP7* and *TRPC6* genes showed significant association in the excessive vomiting meta-analysis. These loci are the second and third most significant association reported by Fejzo et al. [27], closely following *GDF15*, which is also replicated in our analysis. It is also important to mention that we managed to replicate 6 out of 6 variants for PE in the HP GWAS results (with all 6 reaching significance in FG). The replicated associations include, in addition to the genome-wide significant rs35954793 in *FGF5*, rs4769612, rs1918975, rs10774624, and rs1421085 variants in *FLT1*, *MECOM*, *SH2B3*, and *FTO* genes, respectively. Two of the aforementioned variants were also replicated in the GH meta-analysis. This result not only highlights the relevance of these markers for PE, but also emphasizes the high relevance of genetic associations identified in the HP meta-analysis. For GD, additional replicated association corresponded to the rs7754840 variant in the *CDKAL1* gene. This locus has also been found in a recent larger-scale meta-analysis [35].

## 4. Discussion

Serious pregnancy complications affect around 15% of all pregnant women. Preeclampsia, gestational diabetes, and preterm birth are among the most common disorders of pregnancy. Multiple studies have been performed to elucidate the genetic susceptibility to pregnancy complications [14]. In this work, we aggregated genome-wide association data from two major public resources, FinnGen and UK Biobank, to identify novel risk loci for pregnancy-related disorders and replicate known genetic associations for these traits. Such an aggregated analysis of data obtained from different ethnic groups not only provides additional statistical support for the identified loci, but is also useful to discover universal genetic markers of complex diseases that are relevant for multiple populations.

It is important to note that the association signal we observed in our analysis mostly comes from the FinnGen dataset. For example, 11 loci were identified as genome-wide significant in FG (compared to only 4 in UKB [14]). In addition, for the loci showing genome-wide significance in the meta-analysis, the significance is largely driven by the FG cohort. The most likely source of such bias is the much greater sample size in FG (Appendix A), though variation in diagnosis definition (especially for traits that are not defined using ICD-10 codes) may also contribute to the problem.

Our analysis allowed us to characterize genome-wide significant risk loci for hypertension in pregnancy, gestational diabetes, excessive vomiting in pregnancy, and preterm birth. Hypertension in pregnancy increases risk of PE and low birthweight [37]. Moreover, variants that are associated with hypertension contribute to risk of PE [11]. In our analysis, we characterized several loci associated with hypertension complicating pregnancy, childbirth, and puerperium (HP) (Figure 1a and Figure 2a), HP likely comprises PE, pre-existing, and pregnancy-induced hypertension; at the same time, we show that: (i) loci with significant genome-wide association with HP overlap with known risk loci for PE; (ii) many known PE risk variants are replicated in the HP GWAS (Table 3); and (iii) genetic association with HP correlates with gene expression in cervix and endocervix. Taken together, our results suggest that loci that contribute to HP according to the data sources used in the study also may contribute to PE. In accordance with this assumption, all loci showing genome-wide significance in the HP analysis are significant at p<0.001 in the GH analysis with a combined FG + UKB cohort, corroborating their specific relevance to pregnancy-induced hypertensive conditions.

In both FG alone and UKB + FG analysis, we observed significant association of the *FGF5* locus with HP (Figure 1a and Figure 2a). The *FGF5* gene is one of the well-characterized PE genes that has previously been implicated in pregnancy complications [38]. *FGF5* was identified as a hypertension-associated gene that influences PE in Central Asian women [11]. In addition to *FGF5*, two more HP risk loci were identified in both FG and the meta-analysis: *PLCE1* and *RGL3*. The *PLCE1* gene encodes a phospholipase C epsilon 1, a protein that was shown to affect migration and differentiation of podocytes [39]. The *RGL3* gene, on the other hand, encodes a nucleotide exchange factor for the Ral GTPase [40]. Both of these loci were previously identified in large-scale association analyses of systolic blood pressure [41]. Furthermore, both *PLCE1* and *RGL3* have recently been identified as PE risk loci in a larger meta-analysis including FG [26].

For GD, our analysis highlighted two key loci showing great significance in both FG and combined FG + UKB analysis. The genes at these loci, *MTNR1B* and *TCF7L2*, are well-characterized and associated with both GD and T2D [42]. The association of *MTNR1B* with GD has also been confirmed in poorly studied populations such as Russian [43]. It is important to note, however, that both *TCF7L2* and *MTNR1B* variants had very low levels of significance in the UKB cohort, which may have resulted from low sample sizes in UKB. Out of the other known GD variants [42], only rs780094 in the *GCKR* gene reached genome-wide significance. This gene encodes a glucokinase regulatory protein that regulates the activity of *GCK*, a key enzyme of glucose metabolism. Both *GCK* and *GCKR* are implicated in various forms of diabetes (reviewed by Li et al. [44]). When FG alone was considered, we also observed the association of the genes in the MHC region with GD. This result is expected, given a large role of immunity in GD [45] and the high degree of pleiotropy of the HLA genes [46].

Our analysis also allowed us to replicate several recently reported genome-wide associations for HG and PTB. Noteworthy, we successfully replicated all three major loci associated with HG using the meta-analysis results for the excessive vomiting in pregnancy GWAS. The replicated loci include the top association reported by Fejzo et al. [27], corresponding to the *GDF15* gene. This gene has the highest expression in the trophoblast cells of the placenta [47]; its product is actively secreted and is believed to be involved in the regulation of appetite [48]. The other replicated associations involve another appetite regulator, *IGFBP7*, and *TRPC6*. HG is the most severe form of nausea and vomiting in pregnancy and occurs in 0.3–2% of pregnancies [49]. Successful replication of HG loci in the EV GWAS with both single-source (FG-only) and combined cohort suggests that the majority of cases diagnosed with EV in the UKB and FG cohorts correspond to HG.

Our meta-analysis allowed us to reproduce a genome-wide association of the *EBF1* locus with PTB (Figure 2c, Table 2). This locus was reported as the most significant for both PTB and gestational duration by Zhang et al. [13]; and the role of *EBF1* in PTB has been extensively studied [28,50]. *EBF1* encodes a transcription factor that regulates B-cell development [51]. Furthermore, the *EBF1* locus is implicated in the regulation of blood pressure [52], suggesting another possible link between PTB and hypertension. The observed genome-wide association of the *EBF1* locus with PTB in the combined FG + UKB meta-analysis highlights the peculiar power of our approach, as the corresponding locus showed statistical significance in neither UKB or FG (Table 2).

Finally, we performed a large-scale genome-wide genetic correlation analysis for pregnancy complications. One of the benefits of using genome-wide association data provided by the biobanks is the ability to use a rich arsenal of statistical methods to analyze genetic correlations and causal relationships between the traits. Such an analysis could identify other diseases that share risk loci with pregnancy complications and highlight functional links between pregnancy complications and other traits. While most of the significant correlations identified in the *r_g_* analysis corroborated previous findings and are consistent in both FG-only and FG + UKB analysis, there are several notable findings that should be considered. First, we have identified a set of previously uncharacterized genetic correlations for GD, GH, and HP. For example, we discovered a significant correlation between dizziness/giddiness and gestational hypertension. A possible mechanistic explanation of this correlation implies that dizziness can result from vascular abnormalities due to hypertension. Other intriguing examples of genetic correlations include an association between GD and gastrointestinal disease. Gastrointestinal complications are not common in GD patients. Moreover, top loci associated with GD are not significant for gastrointestinal disease, with the exception of the HLA locus. Hence, the pathogenetic basis of this correlation requires further research, especially given that they did not replicate for the results of meta-analysis. It is also worth noting that our analysis did not provide enough statistical support for correlation between GD and obesity ($r_{g}$ = 0.53 ± 0.3, unadjusted *p*-value = 5.48 × 10${}^{-4}$), while strong correlation was discovered between obesity and HP. These findings may either be the result of a low statistical power to detect correlation when testing a large array of thousands of trait pairs or indicate that genetic predisposition to obesity plays a greater role in hypertensive disorders of pregnancy rather than GD. These hypotheses, however, require further investigation.

## 5. Conclusions

In this work, we leveraged two biobank-scale datasets to identify risk loci for pregnancy complications. We identified a total of 12 genome-wide significant loci, 11 of which showed significance in FinnGen, and 6—when combining the two data sources. In addition, we replicated 14 previously reported genome-wide associations for such traits as PE, GD, HG, and PTB. The identified loci, especially the ones showing significance in the meta-analysis, can be considered as robust genetic markers that might be used for the assessment of risk of pregnancy complications. Furthermore, the identified link between the cervix gene expression and pregnancy hypertension suggests that the uterus may play a greater role in the development of hypertensive disorders of pregnancy. We believe that both biobank data and trans-biobank genetic research are an important resource that can be used to highlight loci that show universal significance across the populations. Genome-wide research in pregnancy complications is, in turn, relevant for enhancing understanding of the molecular pathways that are involved in the pathogenesis of pregnancy-related disorders, and can provide valuable information about pathological mechanisms behind these traits.

## Figures and Tables

**Figure 1 genes-13-02255-f001:**
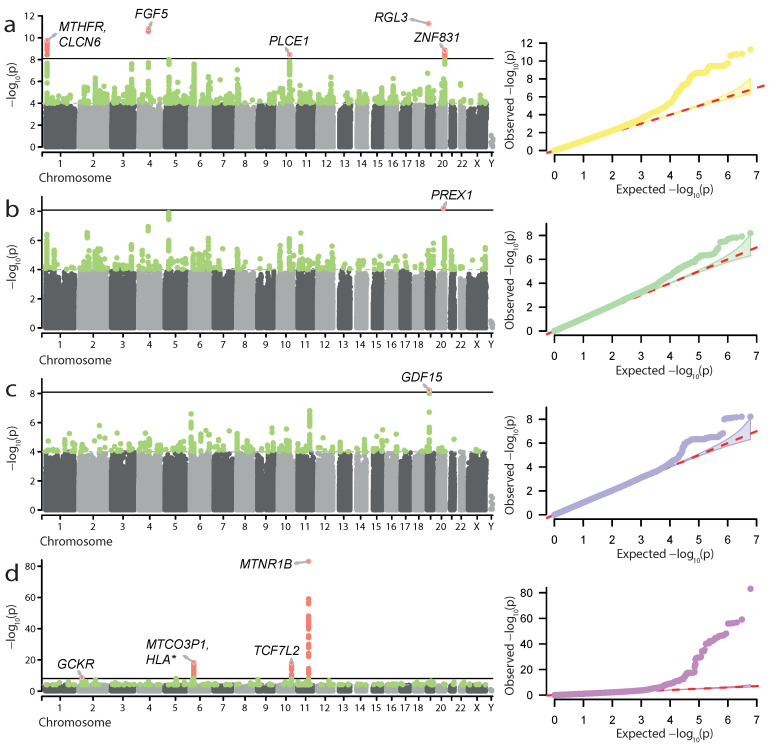
Genome-wide association results for hypertension complicating pregnancy, childbirth, and the puerperium (**a**), gestational hypertension (**b**), excessive vomiting in pregnancy (**c**), and gestational diabetes (**d**) in the FinnGen data. Manhattan plots and quantile-quantile plots are shown. Significant loci and lead SNP genes are highlighted.

**Figure 2 genes-13-02255-f002:**
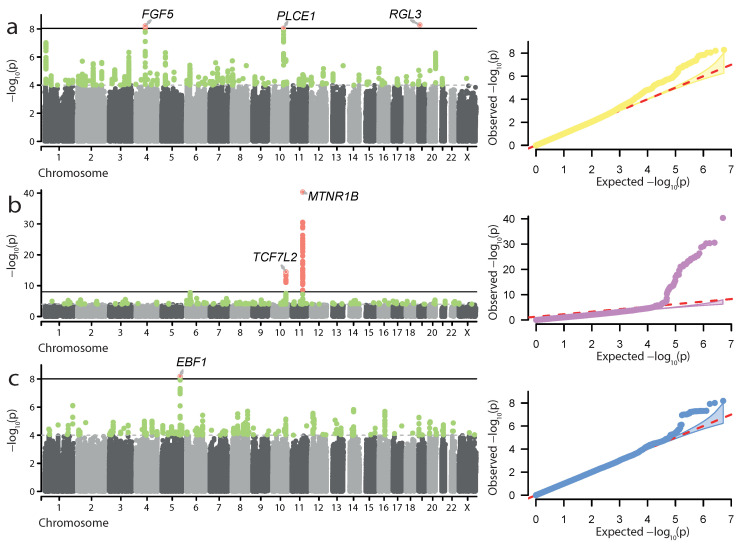
Genome-wide meta-analysis results for hypertension complicating pregnancy, childbirth, and the puerperium (**a**), gestational diabetes (**b**), and preterm birth (**c**). Manhattan plots and quantile-quantile plots are shown. Significant loci and lead SNP genes are highlighted.

**Figure 3 genes-13-02255-f003:**
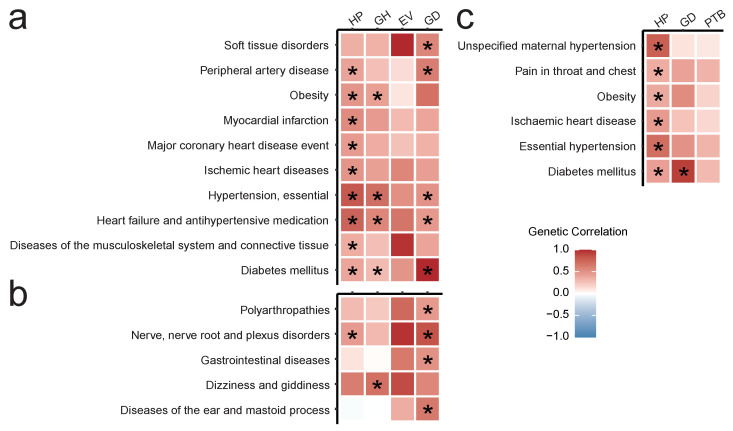
Heatmaps showing: (**a**,**b**) Significant genetic correlations in the FG dataset for hypertension complicating pregnancy, childbirth, and the puerperium (HP), gestational hypertension (GH), excessive vomiting in pregnancy (EV), and gestational diabetes (GD): (**a**) supported by other studies and/or mechanistic evidence, (**b**) novel or not previously reported; (**c**) significant genetic correlations of HP, GD, and preterm birth (PTB) computed using UKB + FG meta-analyses collection. Star sign represents significant correlation in the Wald test after FDR correction. Correlation limits are set to (−1, 1).

**Table 1 genes-13-02255-t001:** Significant genome-wide associations for four pregnancy complications in FinnGen (FG) data.

Trait	Lead Variant	rsID	Gene	Effect	*p*-Value	Meta *p*-Value
HP	1:11865804:A>G	rs13306561	*MTHFR*	intron variant	$1.8\times 10^{-10}$	$9.5\times 10^{-8}$
	4:81188513:C>A	rs35954793	*FGF5*	intron variant	$1.6\times 10^{-11}$	$6.1\times 10^{-9}$
	10:95892788:T>A	rs10882398 ^†^	*PLCE1*	intron variant	$3.5\times 10^{-9}$	$8.9\times 10^{-9}$
	19:11526765:T>G	rs167479 ^†^	*RGL3*	missense variant	$5.1\times 10^{-12}$	$5.2\times 10^{-9}$
	20:59160402:A>C	rs259983	*ZNF831*	intron variant	$1.6\times 10^{-9}$	$8.7\times 10^{-7}$
GH	20:47408414:A>G	rs2208589 ^†^	*PREX1*	intron variant	$6.3\times 10^{-9}$	$2.7\times 10^{-4}$
EV	19:18493064:<del>	rs58835482	*GDF15*	UTR variant	$6.1\times 10^{-9}$	n.a. **
GD	2:27741237:T>C	rs780094	*GCKR*	intron variant	$2.7\times 10^{-9}$	$4.0\times 10^{-6}$
	6:32668411:G>A	rs9275373	HLA *	n.a.	$4.4\times 10^{-19}$	$1.8\times 10^{-8}$
	10:114782581:G>GCT	rs10659211	*TCF7L2*	intron variant	$1.7\times 10^{-18}$	$3.8\times 10^{-15}$
	11:92708710:C>G	rs10830963	*MTNR1B*	intron variant	$7.9\times 10^{-84}$	$4.5\times 10^{-41}$

*—variant corresponds to the MHC region. **—top-variant not present in UKB. †—novel variants (also reported in a parallel FinnGen-including meta-analysis [26]).

**Table 2 genes-13-02255-t002:** Significant genome-wide associations identified in meta-analysis of UK Biobank (UKB) and FinnGen (FG) data.

Trait	Led Variant	rsID	Gene	Effect	FG *p*-Value	UKB *p*-Value	Direction	Meta *p*-Value
HP	4:81188513:C>A	rs35954793	*FGF5*	intron variant	$1.6\times 10^{-11}$	$3.0\times 10^{-2}$	– –	$6.1\times 10^{-9}$
	10:95892788:T>A	rs10882398 ^†^	*PLCE1*	intron variant	$3.5\times 10^{-9}$	$5.5\times 10^{-3}$	– –	$8.9\times 10^{-9}$
	19:11526765:T>G	rs167479 ^†^	*RGL3*	missense variant	$5.1\times 10^{-12}$	$3.8\times 10^{-2}$	++	$5.2\times 10^{-9}$
GD	10:114774433:A>C	rs36090025	*TCF7L2*	intron variant	$1.9\times 10^{-18}$	$2.1\times 10^{-3}$	++	$3.5\times 10^{-15}$
	11:92708710:C>G	rs10830963	*MTNR1B*	intron variant	$7.9\times 10^{-84}$	$1.4\times 10^{-1}$	++	$4.5\times 10^{-41}$
PTB	5:157907974:C>T	rs2963457	*EBF1*	intergenic variant	$1.9\times 10^{-6}$	$2.0\times 10^{-4}$	++	$6.5\times 10^{-9}$

†—novel variants (also reported in a parallel FinnGen-including meta-analysis [26]).

**Table 3 genes-13-02255-t003:** Replication of known associations with pregnancy complications in the FG cohort or combined UKB + FG analysis.

Variant *	rsID	Gene	Trait **	Reference	UKB *p*-Value	FG *p*-Value	Meta *p*-Value
4:57484899:<del>	rs143409503	*IGFBP7*	HG	[27,36]	$1.8\times 10^{-1}$	$2.5\times 10^{-5}$	$2.2\times 10^{-4}$
11:101390067:G>A	rs2508362	*TRPC6*		[27]	$3.9\times 10^{-1}$	$3.1\times 10^{-7}$	$1.1\times 10^{-4}$
13:28564361:C>T	rs4769612	*FLT1*	PE	[36]	$2.2\times 10^{-2}$	$2.3\times 10^{-3}$	$3.1\times 10^{-4}$
3:169462088:C>A	rs1918975 ^†^	*MECOM*		[11,36]	$7.2\times 10^{-2}$	$4.8\times 10^{-5}$	$1.1\times 10^{-4}$
12:111395984:G>A	rs10774624	*SH2B3*			$7.4\times 10^{-1}$	$4.1\times 10^{-7}$	$7.1\times 10^{-4}$
16:53767042:T>C	rs1421085	*FTO*			$9.7\times 10^{-1}$	$1.4\times 10^{-5}$	$6.6\times 10^{-3}$
6:20661019:G>C	rs7754840	*CDKAL1*	GD	[12,36]	$5.6\times 10^{-1}$	$1.3\times 10^{-6}$	$8.4\times 10^{-3}$

*—shown are variants that do not belong to genome-wide significant loci in Table 1 and Table 2 and are significant in meta-analysis (*p* < 0.0012); **—replication of preeclampsia-related associations was performed in the HP meta-analysis results, replication of HG associations was performed in the EV meta-analysis; †—indicated variants are also significant in the GH meta-analysis.

## Data Availability

All data and code pertinent to the results presented in this work are available at https://github.com/bioinf/pregnancy_meta_analysis/).

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
