# Peer review of "Aggregation of Genome-Wide Association Data from FinnGen and UK Biobank Replicates Multiple Risk Loci for Pregnancy Complications"

_genes, 2022, doi:10.3390/genes13122255_

Round 1

Reviewer 1 Report

This study provides GWAS summary statistics from two publicly available biobank-scale cohorts. I have the following comments:

1.      In the introduction section the authors should add incidence values for pregnancy complications (lines 19-21) in Russia and worldwide

2.      How could these findings be translated to clinical practice? Would it be prudent to do a pre-pregnancy search for risk loci in women with risk factors?

3.      In the conclusion section the authors should further discuss cost-effectiveness of their statement (‘we believe that trans-biobank genetic research is an important tool for enhancing understanding of the molecular pathways’)? In what way would these findings help to personalise the evaluation of pregnant women with risk factors for some of these complications?

Author Response

This study provides GWAS summary statistics from two publicly available biobank-scale cohorts. I have the following comments:

  1.     In the introduction section the authors should add incidence values for pregnancy complications (lines 19-21) in Russia and worldwide

Authors: We thank the Reviewer for this suggestion. The corresponding values have been added to the manuscript (lines 22-27).

  1.     How could these findings be translated to clinical practice? Would it be prudent to do a pre-pregnancy search for risk loci in women with risk factors?

Authors: We thank the Reviewer for this question. We added several sentences discussing the opportunities for results’ application to the Conclusion section discussing the development of risk scores based on the identified robust associations, as well as future research prospects.

  1.     In the conclusion section the authors should further discuss cost-effectiveness of their statement (‘we believe that trans-biobank genetic research is an important tool for enhancing understanding of the molecular pathways’)? 

Authors: In fact, the cost-effectiveness of the trans-biobank studies is not a major issue as such studies are rarely conducted from scratch - usually, cross-biobank meta-analyses are performed using data that are routinely collected by the local biobanks and other projects. Hence, it does not require additional resources (besides computing), while providing important insights into the robustness of the genetic associations identified. 

In what way would these findings help to personalise the evaluation of pregnant women with risk factors for some of these complications?

Authors: We have expanded the Conclusions section of the manuscript to consider the application of the results in personalized medicine.

Reviewer 2 Report

In this article, Changalidis and colleagues leveraged two large biobank datasets to conduct a genome-wide association study on multiple pregnancy complications. Extending from the authors’ prior work on pregnancy complications in UKB, they further retrieved FinnGen summary statistics, conducted functional enrichment of the top hits, calculated genetic correlation with other traits, and meta-analyzed FinnGen with UKB. They reported genome-wide significant associations with pregnancy hypertension, gestational diabetes, and preterm birth. 14 out of 40 previously associated loci were replicated in their study. 

My comments are listed before.

1.     The authors reported four genome-wide significant loci in their previous UKB study. By comparing table 1 from that previous study and table s2 from the current study, none of the four UKB loci were supported by FinnGen. This may suggest that the two datasets were different, for example in the definition of diagnosis, and were not appropriate to combine to generate a meta-analysis. This discrepancy hampered both the previous and the current study. 

2.     The fine-mapping of the associated GWAS loci was not complete and left much to be done. In tables and supplementary tables, I assume the genes reported were the nearest gene of the lead SNP (which needs an explicit explanation in the main text). The human genetics field now has better means to assign putatively functional genes underlying GWAS signals than simply relying on physical proximity. There are various fine mapping tools, such as FINEMAP, that the authors could leverage to provide more in-depth knowledge on the putative causal genes and the pathogenesis of pregnancy complications. It would be a missed opportunity to let go of this low-hanging fruit.

3.     Please expand on Methods to include more necessary details. For example, what covariates have been adjusted for in the original UKB and FinnGen GWAS? What was the p-value threshold used for filtering genes as inputs for the FUMA analysis? FUMA did not identify any significant geneset, was that because only genome-wide significant SNPs were included? What was the significant p-value threshold used in the rg analysis? What was the threshold used for replicating previous associations? How were they determined?

4.     Two versions of summary statistics were used as input when calculate rg with other traits: FinnGen alone and FinnGen-UKB meta. How were the results different and overlapped, and how do the authors explain that?

5.     The finding that gestational diabetes and diabetes have an rg=1 is interesting. Furthermore, the authors did not observe a significant rg between gestational diabetes and obesity, which is known to be genetically correlated with diabetes. What do these results mean and what are the implications? The Discussion could be expanded on this point.

6.     It would be much helpful if the authors could provide a contextualized introduction on the three distinct diagnoses analyzed: preeclampsia, HP, and GH. How are they different and overlapping each other?

7.     Line 151. Please show the data underlying the claim that genes specifically expressed in cervix and endometrium and blood were enriched. 

8.     Please clearly indicate in the tables and main text which associations are first identified in this study and which have been implicated before. 

9.     Add chromosome numbers to table 1-3. 

10.  Line 161-165 does not belong in Results. 

11.  Line 251, a partial overlapping between GWAS loci in HP and PE should not be taken to mean that HP loci are specific to pregnancy. Pre-existing hypertension also contributes to HP.

Author Response

In this article, Changalidis and colleagues leveraged two large biobank datasets to conduct a genome-wide association study on multiple pregnancy complications. Extending from the authors’ prior work on pregnancy complications in UKB, they further retrieved FinnGen summary statistics, conducted functional enrichment of the top hits, calculated genetic correlation with other traits, and meta-analyzed FinnGen with UKB. They reported genome-wide significant associations with pregnancy hypertension, gestational diabetes, and preterm birth. 14 out of 40 previously associated loci were replicated in their study.  

Authors: We thank the Reviewer for a thorough assessment of our work and useful comments and suggestions. Please find our point-by-point responses below.

My comments are listed before.

  1.     The authors reported four genome-wide significant loci in their previous UKB study. By comparing table 1 from that previous study and table s2 from the current study, none of the four UKB loci were supported by FinnGen. This may suggest that the two datasets were different, for example in the definition of diagnosis, and were not appropriate to combine to generate a meta-analysis. This discrepancy hampered both the previous and the current study.

Authors: We agree with the Reviewer that the two datasets might have some systematic difference. However, as shown in Supplementary Table S1, the majority of phenotypes included into the analysis are reported using the ICD-10 codes. Hence, we believe that the difference in diagnosis is not the major reason for the fact that meta-analysis does not show sufficient levels of replication. To the contrary, we believe that most of the differences come from differences in sample size, which are already discussed in the manuscript. We added an additional paragraph pointing out the difference to the Discussion section (lines 273-279).

  1.     The fine-mapping of the associated GWAS loci was not complete and left much to be done. In tables and supplementary tables, I assume the genes reported were the nearest gene of the lead SNP (which needs an explicit explanation in the main text). The human genetics field now has better means to assign putatively functional genes underlying GWAS signals than simply relying on physical proximity. There are various fine mapping tools, such as FINEMAP, that the authors could leverage to provide more in-depth knowledge on the putative causal genes and the pathogenesis of pregnancy complications. It would be a missed opportunity to let go of this low-hanging fruit.

Authors: We added the results of statistical fine-mapping to our manuscript. To this end, we acquired fine-mapping results from the FinnGen portal, as well as performed fine-mapping of GWAS signal in the loci significant in the meta-analysis. These results are now shown in the corresponding parts of the Results section (lines 150-156, 175-179). As can be seen from the text and tables, the vast majority of potentially causal variants are still located in the non-coding regions, with rare exceptions that are now discussed in the text.

  1.     Please expand on Methods to include more necessary details. For example, what covariates have been adjusted for in the original UKB and FinnGen GWAS? What was the p-value threshold used for filtering genes as inputs for the FUMA analysis? FUMA did not identify any significant geneset, was that because only genome-wide significant SNPs were included? What was the significant p-value threshold used in the rg analysis? What was the threshold used for replicating previous associations? How were they determined?

Authors: We added more information about the corresponding parameters and dataset properties to the Methods (lines 81-88). As for the FUMA analysis, significant genes were identified in the majority of the datasets that were annotated - these results, however, were omitted when preparing the manuscript draft due to the fact that no genes other than the lead SNP genes were identified in the MAGMA gene analysis performed in FUMA. We have now included all FUMA results as the Supplementary File for the reader’s convenience.

  1.     Two versions of summary statistics were used as input when calculate rg with other traits: FinnGen alone and FinnGen-UKB meta. How were the results different and overlapped, and how do the authors explain that?

Authors: As can be seen from Figure 3 and the corresponding text, three major groups of traits (hypertension and cardiological disorders, diabetes, and obesity) showed significant correlation with HP and GD in both FG. and FG+UKB analysis. As for the novel correlations, predominantly identified for GD in the FG data - these findings are not reproduced in the FG+UKB analysis at the required level of statistical significance. We added a sentence discussing this issue to the Discussion section (lines 339-362). This result, however, may be explained either by insufficient statistical power resulting from low GD case count in UKB. 

  1.     The finding that gestational diabetes and diabetes have an rg=1 is interesting. Furthermore, the authors did not observe a significant rg between gestational diabetes and obesity, which is known to be genetically correlated with diabetes. What do these results mean and what are the implications? The Discussion could be expanded on this point.

Authors: We thank the Reviewer for this comment. When analyzing the results, we also paid attention to the details mentioned. For the GDM-DM correlation, we believe that the exact value of 1 is an overestimate by the toolkit used for analysis. The actual confidence interval of the estimate is 0.72 - 1.42, and we believe that the true value should be closer to the lower boundary. We added the errors of the estimates to the Results section to enhance the reader’s understanding. As for the lack of genetic correlation between obesity and GDM, the results show a high estimated rg value (rg = 0.53±0.3, unadjusted p-value = 5.48 ✕ 10-4), but the error of the estimate is too large to make the estimate significant after FDR in this particular case. We now discuss the issue in the Discussion section (lines 353-360)

  1.     It would be much helpful if the authors could provide a contextualized introduction on the three distinct diagnoses analyzed: preeclampsia, HP, and GH. How are they different and overlapping each other?

Authors: We added more details regarding the definitions of the traits and sample overlap to the Methods section. (lines 67-80).

  1.     Line 151. Please show the data underlying the claim that genes specifically expressed in cervix and endometrium and blood were enriched.

Authors: We added a new table to the Supplementary Information showing the results of the gene expression enrichment analysis from FUMA.

  1.     Please clearly indicate in the tables and main text which associations are first identified in this study and which have been implicated before. 

Authors: We have added the necessary details to the text and tables. As indicated in the manuscript, the majority of associations are known. The only novel findings correspond to HP/GH loci; however, it is important to indicate that the novel associations have also been recently identified in a parallel study that also included the FinnGen cohort (and is available only as a medRxiv preprint: https://doi.org/10.1101/2022.05.19.22275002). Nevertheless, we believe that an independent validation of these associations using our FG+UKB meta-analysis is an important step towards characterizing robust risk loci for the traits under consideration.

  1.     Add chromosome numbers to table 1-3. 
  2. Line 161-165 does not belong in Results.

Authors: Issues (9-10) were corrected.  Contents of all tables have been expanded.

  1. Line 251, a partial overlapping between GWAS loci in HP and PE should not be taken to mean that HP loci are specific to pregnancy. Pre-existing hypertension also contributes 

Authors: We agree that the wording we used in the original draft is not totally correct. We have made the necessary corrections in the Discussion (lines 285-291). 

Round 2

Reviewer 2 Report

The authors have satisfactorily addressed my original concerns. The authors, though, need to update their abstract, which seemed to have been left unchanged inadvertently during the revision. Figures mentioned in the abstract do not align with that in the latest version of the conclusion (e.g., 6 vs 7 genome-wide significant loci), and some new results and conclusions were not mentioned at all in their abstract (e.g. implication of cervix gene expression in pregnancy hypertension ). 

Author Response

The authors have satisfactorily addressed my original concerns. The authors, though, need to update their abstract, which seemed to have been left unchanged inadvertently during the revision. Figures mentioned in the abstract do not align with that in the latest version of the conclusion (e.g., 6 vs 7 genome-wide significant loci), and some new results and conclusions were not mentioned at all in their abstract (e.g. implication of cervix gene expression in pregnancy hypertension ). 

Authors: The corresponding changes have been made to the abstract and conclusions.